# Comparing the Environmental Impacts of Meatless and Meat-Containing Meals in the United States

**Alexi Ernstoff** [1] , **Qingshi Tu** [2] , **Mireille Faist** [3] , **Andrea Del Duce** [3,4] , **Sarah Mandlebaum** [2] and **Jon Dettling** [2],*

[1] Quantis, EPFL Innovation Park-Bâtiment D, 1015 Lausanne, Switzerland; Alexi.Ernstoff@Quantis-intl.com

[2] Quantis, 240 Commercial Street #3B, Boston, MA 02109, USA; qingshi.tu@yale.edu (Q.T.); sarah.mandlebaum@yale.edu (S.M.)

[3] Quantis, Konradstrasse 52, 8005 Zürich, Switzerland; mireille.faist@quantis-intl.com (M.F.); andrea.delduce@zhaw.ch (A.D.D.)

[4] Institute of Sustainable Development, Zurich University of Applied Sciences, 8401 Winterthur, Switzerland

\* Correspondence: jon.dettling@quantis-intl.com

**Abstract:** This study compares the environmental impacts of meatless and meat-containing meals in the United States according to consumption data in order to identify commercial opportunities to lower environmental impacts of meals. Average consumption of meal types (breakfast, lunch, dinner) were assessed using life cycle assessment. Retail and consumer wastes, and weight losses and gains through cooking, were used to adjust the consumption quantities to production quantities. On average, meatless meals had more than a 40% reduction in environmental impacts than meat-containing meals for any of the assessed indicators (carbon footprint, water use, resource consumption, health impacts of pollution, and ecosystem quality). At maximum and minimum for carbon footprint, meat-containing dinners were associated with 5 kgCO$_2$e and meatless lunches 1 kg CO$_2$e. Results indicate that, on average in the US, meatless meals lessen environmental impacts in comparison to meat-containing meals; however, animal products (i.e., dairy) in meatless meals also had a substantial impact. Findings suggest that industrial interventions focusing on low-impact meat substitutes for dinners and thereafter lunches, and low-impact dairy substitutes for breakfasts, offer large opportunities for improving the environmental performance of the average diet.

**Keywords:** diets; life cycle assessment; vegetarian

## 1. Introduction

Consuming animal products is a well-recognized driver of greenhouse gas emissions, water consumption, biodiversity loss, and nutrient cycling impacts due to agricultural production [1]. It is predicted that technological advancements in livestock production practices have a limited ability to decrease environmental impacts—for example, due to the low biomass conversion efficiency of cattle and limited possible improvement in nitrogen fertilizer efficiency for animal feed [2–4]. Thus decreasing meat and animal product consumption (shifting diets) to decrease demand and production quantity is recommended as a key environmental impact mitigation strategy [5,6]. In order to quantify the environmental benefit of reducing animal product consumption, research investigating the environmental impacts of food systems considers various perspectives, for example, relatively ranking individual food items [7] or dietary patterns (such as typical versus a vegetarian diet) [8].

When comparing food items, environmental impacts are often compared per mass (e.g., kilogram), serving size, energy (e.g., kilocalorie), or protein [5,7,9]. Knowing the relative ranking of the environmental impacts of various foods has been considered important to identify food groups of

concern and to develop a basic scientific understanding of the impacts of food systems. Relative ranking of food items also demonstrates the complexity of assessing the impacts of food consumption, as using different metrics—e.g., greenhouse gas (GHG) emissions per kilogram, kilocalorie, gram protein or servings—can change the relative ranking of food items [7,10,11]. Relative ranking can also help understand the differences between local and imported foods [12], and differences between geographical settings and farming practices—although such differences are notoriously challenging to capture using most widely available quantitative sustainability assessment methods [13–15]. Further adding to complexity when interpreting food-to-food comparisons to assess diets, comparing across food products does not consider the context of consumption patterns; for example, impacts per kilogram of a selected food may be substantially higher, but could be typically consumed in lesser amounts. The relative ranking across food items therefore does not provide enough information to guide informative decision-making to recommend dietary changes.

Given that relative ranking between food items does not always clearly help inform dietary shifts, comparing the environmental impacts of diets (i.e., what is consumed on average per day), has become another common way of assessing the environmental impacts of food consumption [7,8,16,17]. Detailed data (e.g., average quantities of food types consumed per day in a country) on different dietary patterns such as vegan (no animal products), vegetarian (no meat products), and even the average diet are largely missing. Many studies assessing diets thereby investigate "constructed" diets, for example, by evaluating regulatory recommendations (i.e., food pyramids) or by making assumptions about the replacement of food items to model a hypothetical diet such as a vegan or vegetarian diet [5,8,18–26]. Assessing hypothetical diets such as food pyramids can help understand the impact of recommended diets, and can offer the range of impacts for hypothetical substitutions [10,27,28]. This can help understand the potential range of impacts for "vegan" and "vegetarian" diets. To reach a wider proportion of the population beyond vegetarians and vegans, many stakeholders are investigating targeting meals (instead of overall diets) as an effective strategy to decrease meat consumption, for example, through mobile phone applications, menu designs, recipes, or meat-replacement products [29–32].

To complement recent market growth with respect to plant-based meat alternatives and the opportunity to implement interventions on a meal-to-meal basis instead of an overall diet basis, we chose to explore meal-to-meal comparisons in this study. Other recent work has also focused on meals, for example, comparing Spanish meals and different meal preparation methods [33–35]. To our knowledge, this is the first meal-to-meal environmental comparison study that uses United States (US) dietary survey data to represent average meal composition. In this study, we focus on the US, a country with meat consumption more than three times the global average [36], and with intake data available to estimate meal compositions and assess respective environmental impacts. Specifically, we used the National Health and Nutrition Examination Survey 2011–2012 (NHANES) [37], the United States Department of Agriculture (USDA) Economic Research Service (ERS) [38], the National Marine Fisheries Service (NMFS) [39], and the USDA Agricultural Research Service (ARS) National Nutrient Database [40]. We used a life cycle assessment (LCA)-based approach to include impacts related to the entire food provision system from primary production to disposal. The primary aim of this study was to gather evidence that reflects realistic consumption quantities and understand whether (and how much) of an environmental benefit can be obtained on a per-event basis if Americans begin to shift towards meatless breakfast, lunch, and/or dinner. The purpose of this study was not to investigate market consequences as demand shifts to a critical level.

## 2. Materials and Methods

We compared different meal types reported in the National Health and Nutrition Examination Survey (NHANES) [37] with the functional unit of providing a US adult with one meal at their home. A meal was defined as a self-reported in-home consumed breakfast, lunch, or dinner occasion and did not include foods consumed between meals or out-of-home. To inform the environmental impact

assessment, we began with meal compositions (quantity in mass of food categories consumed on average daily), and the life cycles of the food categories within the meals were then constructed. The life cycle of all considered food categories consists of production (harvest quantity), food manufacturing, packaging, distribution (refrigeration and transport), consumer use (storage, preparation, and cleaning), and waste (retail and consumer).

### 2.1. Meal Compositions, Food Groups and Sub-categories

The average compositions of meat-containing and meatless meals were first determined from NHANES [37]. NHANES 2011–2012 (the most recent version at the time this study was conducted) consists of self-reported 24-hour dietary recall survey data from 4948 male and female adults (19+ years). NHANES data were used to identify the food groups consumed during breakfasts, lunches, and dinners in the United States, and their relative proportions. The sum of average breakfast, lunch, and dinner meals does not equate to average daily intake as out-of-mealtime consumptions (e.g., snacks) were not included.

We constructed average meal compositions based on key food groups and subcategories, to synthesize the array of food items in the NHANES data and simplify the assessment. The food groups were: dairy (milk drinks and other dairy); meats (including fish and any meat mixtures); eggs (including egg substitutes such as whites and powders); vegetables (including potatoes); legumes, nuts, and seeds; fruits and fruit juices; fats, oils, and dressing; and sugars and sweets. These food groups and the corresponding food subcategories are listed in the first column of SI Table S1. The average consumption of each subcategory and overall food group during each meal occasion was quantified, the latter as a necessary pre-step to quantify the environmental impacts, and the former as a way to visualize the consumption of various categories and to match to food loss and waste categories. A meat-containing meal included any meal that contained any amount of food that corresponded to one of the groups of meat, poultry, or fish, or any meat mixtures.

For the meat mixtures, equal proportions of the specified mixtures were assumed. For example, meat and vegetable mixtures were assumed to contain half "vegetable" and half "meat" by weight, and "meat", "grain", and "vegetable" mixtures were assumed to contain one-third of each of these food groups. A meat mixture was modeled as a combination of beef, chicken, and pork, and fish, proportional to the consumptions in 2012 as reported in USDA ERS [38] and NMFS [39], and listed in SI Table S1. Meals were scaled to ensure that the same quantity of food was consumed in both meat-containing and meatless meals. In the meal composition, fluid milk and fruit juices were considered, but water, soda and other sweetened beverages were excluded assuming that consumption of these products alongside meals does not vary greatly between meatless and meat-containing options. If they were included, these other beverages would be the majority of the food weight consumed.

The composition of meatless meals should not be interpreted as reflecting the dietary patterns of vegetarians, as non-vegetarians also consume meatless meals. The meals should be interpreted as the average meal consumptions for US adults in 2011–2012.

### 2.2. From Composition to Life Cycle Considerations

In this study, we began with consumption data (for example, of cooked food) to obtain the composition of each meal. From the composition, we then estimated the quantity of raw food produced, and constructed the life cycle processes (e.g., production, manufacturing, packaging, transport, preparation, clean-up, and waste associated with the foods in the meals).

Agricultural production of food is typically the largest contributor to environmental impacts across the life cycle [41]. We therefore did not perform a detailed review of the other life cycle considerations (i.e., packaging, cooking, etc.) for each food item in the meal types, and instead included these aspects as generic estimates. To obtain the raw quantity of food produced, adjustments were made to account for retail and consumer wastes, as well as water-weight lost or added during cooking. Food wastes were considered per food group and, in some cases, per subcategory when data were

available [42], and were assumed to be equal for meatless and meat-containing meals (that is, the food waste percentage for each category was assumed to be independent of the meal occasions and meal types). The considered waste values are presented in SI Table S2, and were simply used as scaling factors such that the amount of food assumed to be produced was equal to the total of the amount consumed plus the wasted amount. Additionally, USDA ARS [40] data were used to convert cooked ingredients to their raw form, again to adjust for lost or gained water-weight during the cooking process and to back-calculate the original quantity. For example, per 1 kg of cooked beans, only 400 g of dry beans are required (the rest is water weight), and therefore the 400 g is used as the reference value for the environmental impact calculations, and not the full 1 kg, which would over-estimate results by more than a factor of 2. All food wastes were assumed to be sent to landfill, which is the primary destination of food waste (about 95%) in the United States and the worst-case scenario.

To elucidate approximate proportional contribution of manufacturing and distribution life cycle stages to the overall impact of meatless and meat-containing meals, we used the same set of assumptions for all meal types. In order to derive the assumption on how much energy is used due to food manufacturing and distribution for an average meal, we used an input-output database. Input-output databases can provide useful information to make assumptions on average values, e.g., on energy consumptions for a given stage in a given sector for each dollar spent in that sector. We used the economic input-output LCA database (EIOLCA) through Carnegie Mellon University [43] to estimate the energy requirements of food manufacturing and distribution separately per meal, given a $7.19 average meal cost [44]. From this starting value, EIOLCA estimates that each dollar spent in the food sector results in between 0.05 and 0.1 MJ of total energy used in the manufacturing stage, and 0.002 MJ in distribution operations (e.g., retail). We thus assumed roughly 0.7 MJ of energy ($7/meal multiplied by 0.1 MJ/$) to manufacture and 0.014 MJ of energy ($7/meal multiplied by 0.002 MJ/$) to distribute 1 meal. The energy consumption per meal was then matched to an LCI entry for the US electrical grid mix to estimate impacts as a realistic worst-case scenario, instead of separating into various forms of energy combustion. We have no available evidence to justify that the values for manufacturing and distribution would be drastically different between meatless and meat-containing meals, therefore the inclusion of these stages in the study is only to estimate the magnitude of the overall impact and not to compare the impacts between the meal types.

Transportation impacts are included within processes associated with life cycle stages. For example, raw material production includes transport of feed to animal farms and transport from farm to manufacturing facility when relevant. We did not modify the distances and vehicles assumed for transport within the life cycle inventory database used (more information is provided in Section 2.3). Generally, transport is a minor part of the overall impact, and we have no reason to suspect additional transportation assumptions should be different for meat-containing and meatless meals.

The amount of packaging required per meal (also assumed to be the same for all meal types) was estimated starting with national statistics of packaging disposal per year, the quantity of this packaging that is related to food, the number of meals consumed in the US in one year, and then adjusting by the estimated amount snacks and beverages consumed, as they were not included in the meal compositions in this study. It follows that, with 75 million tons of packaging waste in the US [45], approximately two-thirds of this related to food packaging [46], approximately 350 billion meals per year (319 million people consuming 3 meals per day for 365 days per year), and finally adjusting for the roughly 50% of food consumed that is snacks and beverages [47,48], there is roughly 70 g of packaging per meal, which is broken down into specific material categories (plastics, paper, aluminum, etc.), based on the ratios reported by the US Environmental Protection Agency (EPA) [45]. Packaging is recycled and disposed of in municipal waste according to US EPA statistics [49]. Impacts associated with the recycling process itself are included, but there is no benefit allocated for the recovery of recycled materials.

In general, food is stored and prepared in a wide variety of ways, with few statistics available or identified to characterize typical values for different foods or meal types. We therefore used the same set of expert assumptions for each meal (scaled to the meal weight) with respect to the energy,

water, and material consumptions required for storing, cooking, and cleaning (i.e., washing). These assumptions are listed in SI Table S3.

*2.3. Life Cycle Inventory and Impact Assessment*

Life cycle assessment (LCA) is an internationally-recognized quantitative assessment approach that evaluates potential environmental impacts of products and services throughout their "life cycle", beginning with raw material extraction and including all aspects of transportation, manufacturing, use, and end-of-life treatment. Over recent decades, LCA and life cycle thinking has become a principal approach to compare the environmental impacts of various foods and diets [8,16,50,51].

The two main quantitative steps of LCA are life cycle inventory (LCI) and life cycle impact assessment (LCIA). LCI identifies the flows of materials, energy, and substances into and out of each considered food product system, and LCIA characterizes the effect of these flows. In this study, the ecoinvent v3.1 and agri-footprint databases [52] were used to gather the LCI of representative unit processes (with ecoinvent used in priority). The original study backing this article was performed in 2015 with ecoinvent 3.1 and SimaPro 8.0.3 software developed by PRé Consultants. Global processes were prioritized (which account for the market mixes), with US processes (which do not account for imports) used if global processes were unavailable. The full list of LCIs for the food categories is available in SI Table S1. Recycled packaging materials were matched to the most appropriate ecoinvent 3.1 entry for material-specific recycling and municipal solid waste in a sanitary landfill. Food wastes were modelled as municipal solid wastes in a sanitary landfill.

IMPACT 2002+ vQ2.2 [53,54] was employed as the LCIA method. Through IMPACT 2002+, 17 impact categories (midpoints) are aggregated into endpoint categories—human health impact of pollution (e.g., by fine particulate matter), ecosystem quality, resource consumption, water use, and carbon footprint—to indicate environmental damages. More detailed explanations of LCA, LCIA and damage categories are available elsewhere [53–57].

## 3. Results

*3.1. Meal Compositions and Associated Food Quantity Produced*

As a first necessary step to compare meat-containing and meatless meals, the composition of meals as reported to be consumed were obtained from NHANES 2011–2012 [37]. The proportions of food groups within meat-containing and meatless meals (prior to any weight-adjustment or waste-adjustment) are presented in Figure 1. The masses consumed of each food group sub-category in each meal type are reported in SI Table S4, along with respective produced quantities, which are the consumed quantities adjusted by wastes and cooking weights. The relative ranking of food items for produced versus consumed quantities for meat-containing meals are presented in SI Figure S1.

On average, meatless meals contained 270.5, 341.1, and 432.1 g, and meat-containing meals contained 365.6, 411.6, and 496.2 g of total food consumed for breakfasts, lunches and dinners, respectively. As input for the environmental impact assessment, however, the weights of the meatless meals were scaled up to equal those of meat-containing meals. This is to ensure a non-biased comparison between what types of foods are eaten rather than how much of each food is eaten. Furthermore, the reasons why meatless meals contained less overall food weight (e.g., perhaps due to gender or age) on average is not clear from the data we had available for this study. Meatless lunches and dinners contained approximately double the amount of grain-based foods and milk drinks than that of meat-containing meals. Additionally, meatless meals contained more nuts and seeds, legumes (except for breakfast), and fruits, and less fats, oils, dressings, and vegetables. Similar amounts of dairy and sweets were consumed in both meal types.

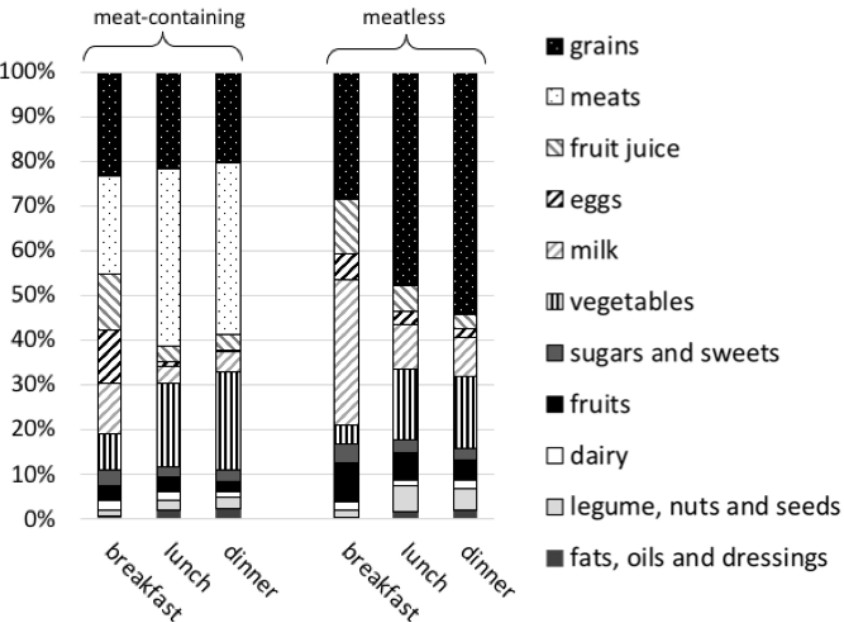

**Figure 1.** Composition of meat-containing and meatless meals by food groups.

## 3.2. Environmental Impact Assessment

### 3.2.1. Contribution of Food Groups

For meat-containing meals, the meat products were the major contributors to all impact categories, despite being only about a third of the mass consumed. In meatless breakfasts, dairy was the largest contributor to carbon footprint, resource consumption, and health impacts of pollution. Grains were the largest contributor to water use. Fruits and fruit juices were the largest contributor to ecosystem quality impacts. (In all cases emissions associated with food production were the majority of the impact). In meatless lunches and dinners, grains tended to be the largest contributor to the overall impact, largely due to the large proportion consumed (per kilogram, grains tend to have a lower impact than other plant-based foods). The relative contribution of different food groups to the overall impact for breakfast is presented in Figure 2, and lunch and dinner are presented in Figures S2 and S3.

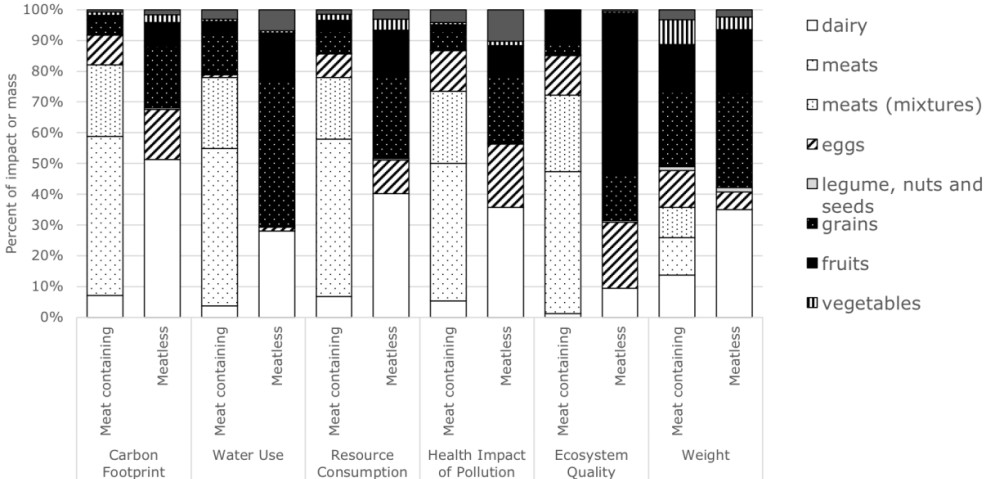

**Figure 2.** Contribution of food categories to the environmental impacts of breakfast (dairy includes milk and fruits includes fruit juices). The relative contribution to the weight of food consumed is demonstrated in the rightmost columns.

### 3.2.2. Impacts of Meal Types and Contribution of Life Cycle Stages

Figure 3 presents the relative environmental impacts comparing meatless to meat-containing meals (scaled to 100%), as well as the contribution of different life cycle stages to the overall impact. The absolute values of the environmental impact indicators are presented in SI Table S5. In summary, the lowest impacting meal with respect to carbon footprint was meatless lunch, with 1 kg $CO_2$e per meal, and the highest impacting meal was dinners containing meat, at 5 kg $CO_2$e per meal. Meatless meals always resulted in less environmental impacts than meat-containing meals, with the biggest advantage for water use and ecosystem quality, and the least advantage for resource consumption (largely due to post-agricultural life cycle stages).

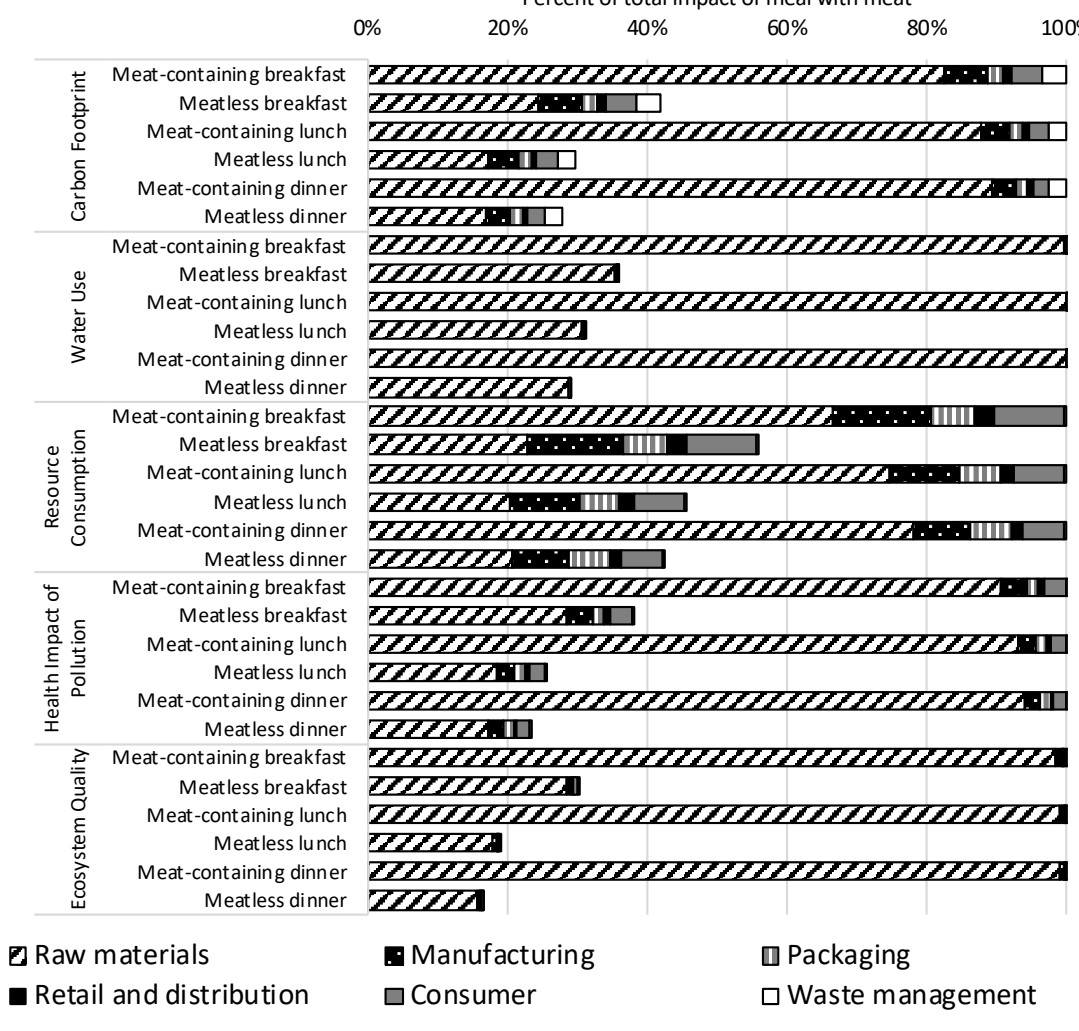

**Figure 3.** Comparing the relative impacts and contributions of lifecycle stages for meatless meals and meat-containing meals for all environmental impact indicators and meal occasions.

For meat-containing meals, agricultural production (raw materials) contributed the largest proportion of environmental impacts in each meal type and represented the majority of the footprint for water use and ecosystem quality. In contrast, subsequent life cycle stages for meatless meals—which had a lower overall footprint—were large contributors to the carbon footprint, resources use, and, to some extent, the health impacts related to pollution.

Overall, the impact reductions ranged from 44% for resource consumption for meatless breakfast to 88% ecosystem quality for meatless dinner. Generally, the benefits for switching to a meatless breakfast are less than those for other meals, reflecting that less meat is, on average, consumed during

breakfast. For meatless breakfasts, dairy (e.g., milk and yogurt) was a predominate contributor to impact, and was consumed approximately 2× more than in meat-containing breakfasts. The impact reductions by percentage and also by absolute value are summarized in Table 1.

**Table 1.** Environmental impact differences for meatless meals compared to meat-containing meals for all meal occasions, with respect to percentage decrease and absolute decrease (impact of meat-containing meal minus impact of meatless meal).

| Meal | Carbon Footprint (kg $CO_2$ eq.) * | | Water Use ($m^3$) * | | Resource Consumption (MJ) * | | Health Impact of Pollution (DALY) * | | Ecosystem Quality (PDF-$m^2$-y) * | |
|---|---|---|---|---|---|---|---|---|---|---|
| | Amt. | % | Amt. | % | Amt. | % | Amt. | % | Amt. | % |
| **Breakfast** | 1.5 | 58% | 0.26 | 64% | 8.3 | 44% | $1.7 \times 10^{-6}$ | 62% | 4.9 | 70% |
| **Lunch** | 2.8 | 74% | 0.56 | 81% | 15.5 | 59% | $3.1 \times 10^{-6}$ | 78% | 9.0 | 85% |
| **Dinner** | 3.7 | 77% | 0.80 | 84% | 20.4 | 64% | $4.0 \times 10^{-6}$ | 81% | 11.6 | 88% |

\* kilogram of carbon dioxide equivalent (kg $CO_2$ eq.); cubic meters ($m^3$); megajoules (MJ); disability-adjusted life year (DALY); potential disappeared fraction of species on a square meter during a year (PDF $m^2$-yr).

### 3.3. Uncertainty Assessment

In general, we found the results of this study to be well-aligned with other LCA-based research where, for example, agricultural production impacts for one day of meals were roughly 5 kg $CO_2e$/day, or nearly 2 tons $CO_2e$/person/year—and up to 4 tons $CO_2$/person/year including all life cycle stages [8,22,58]. In any case, there are several sources of uncertainty that can influence the results in this study, including the composition of the diets from the dietary survey data (NHANES) and the assumption of the mixture compositions (e.g., types of meats in a mixed meat product) described in Section 2.1. Accounting for uncertainty of consumption data is extremely challenging and thus considered outside the scope of this study. However, other work suggests that memory recall data is highly uncertain and should thereby be used with caution [59].

As for mixture assumptions—if the same set of assumptions are used for dish types that occur in both meat-containing and meatless meals—then the influence of these assumptions is most important in the case of mixtures containing meat. The proportion of mixtures with both non-meat and meat items e.g., "vegetables with meat, poultry, fish", was never more than 2% of the total meal mass and thus changing the assumptions would not drastically influence results. Meats as a category and mixtures of meats where the meat type was not known accounted for up to 30% of the mass of meat-containing dinners and thereby the type of meat consumed could have an important influence on the results. To understand the influence of meat type, a sensitivity analysis was performed using a 100% total of beef, pork, or chicken as the meat type in each meal; see SI Table S6. This analysis demonstrated that meat containing meals that are 100% chicken have similar impacts related to the non-meat containing meals with respect to carbon footprint, whereas meals containing 100% beef are substantially higher. This finding underscores the large difference in impacts across meat types and not all "meat-containing" meals should be assumed to be more impacting than meatless meals—especially meatless meals containing other animal products (e.g., dairy). Other recent studies have done a thorough investigation that involves disaggregating food types and have come to similar conclusions as our study with respect to both magnitude of impacts and predominating contributors to US dietary impacts [60,61].

As an additional uncertainty, this study's focus on meals rather than overall daily intake made consistent accounting of beverages a major challenge. Including all beverages (e.g., water or soda) as part of meal occasions is not practical because beverages then dominate the overall mass consumed, which then skews results when scaling meatless meals to the same mass as meat containing meals. Similar issues would occur if scaling diets by protein or by calories, as milk would influence protein consumption and calories would be influenced by sugary sweetened beverages. As a compromise,

fruit juice and milk drinks were included when reported with meal occasions for the following reasons. First, fruit juices and milk drinks have higher environmental impacts than most other beverages (water and sodas, for example). Fruit juices were consumed at relatively low amounts and in similar amounts between meatless and meat containing meals. Milk and milk drinks were additionally included as data demonstrated high consumption of milk drinks accompanying meatless meals, which if neglected would show artificially lower environmental impact of meatless meals in comparison to actual practices, e.g., where milk may accompany cereals in the morning as part of a meal. The uncertainty regarding how to treat beverages when considering meals is tied to the reality that different beverages serve different functions in the diet, which are subjectively "part of a meal" or not. Diet-level studies offer a more holistic vision of the impact of beverages; for example, in Heller et al. (2018) beverages were a non-negligible source of impact in the US diet [60].

Another type of uncertainty that can influence the results of this study is related to the life cycle inventory (LCI) matching and modeling, and life cycle impact assessment (LCIA) modelling. Several issues are that there is no information within dietary surveys regarding food production region, and using another data source (e.g., import and export tables) to match this information to each food category is a substantial undertaking. Furthermore, even with these data available, LCI database entries only cover a fraction of the thousands of crop–country combinations on the market. Global LCI entries were thus chosen in priority in this study to represent a market mix informed by economic and agricultural statistics, and are thus considered a robust indication of impacts for impact screening.

The uncertainty related to food production origin is, however, becoming increasingly important given deforestation, and other types of land use change (which are highly sensitive to the agricultural production region) are being increasingly recognized as a major source of carbon emission and loss of biodiversity. In LCA, such impacts are sensitive to not only variables such as measurable variations in regional soil carbon, but also political decisions regarding time horizons, and market-influences related to consequential market shifts and indirect land use change [61,62]. In this study, the uncertainty related to land use change is highly relevant for meat and non-meat animal products (e.g., eggs and dairy), which may have been produced with feed from regions undergoing extensive deforestation (e.g., Brazil). To further quantify the uncertainty of the LCI and LCIA would require further research and is discussed elsewhere, e.g., the user guide of IMPACT 2002+ available at quantis-intl.com [53].

## 4. Discussion

We found in all cases—for both meal occasions and for each environmental impact indicator—average meatless meals had lower impacts than average meat-containing meals. At maximum and minimum for carbon footprint, meat-containing dinners were roughly 5 $kgCO_2$e and meatless lunches were roughly 1 kg $CO_2$e. Furthermore, since we adjusted meatless meals in overall quantity of food consumed to match meat-containing meals, the impacts of meatless meals would be even more profound if estimated by the reported food quantity. NHANES [37] data were used to estimate average meatless and meat-containing meals to employ a data-driven approach to estimate meal consumptions, instead of subjectively assuming types of meat replacements. Recent studies also assessing US diet (not meal-to-meal) comparisons have similar results, suggesting that animal product production (not just meat production) makes a large contribution to the overall footprint of US diets [1–4]. As for studies outside of the US, similar findings have also been reported, although there are regional variations in consumption habits; for example, much more seafood and thus higher associated impact in Spain [33].

Dietary recall surveys such as NHANES can lead to reporting biases—for example, underestimating energy intake [63], although protein reporting (for example, through meat consumption) may be more accurate [64]. Given limitations of dietary surveys, the environmental impacts estimated for reported average meal compositions are indicative of the magnitude of environmental advantage of meatless meal occasions. Due to data availabilities and challenges in also accounting for demographic differences in meal choices, and the potential that a person may choose to eat less, or less animal products at a given meal and compensate at the next meal occasion, we did not attempt to compare

the "most impacting" meatless meal with the "least impacting" meat-containing meal. Looking at extremes of dietary choices would be more meaningful at the level of diets than at the level of meals, and should also include information on nutrition and consider potential demographic differences.

This study offers no decision-making guidance that requires understanding the implications of uncertainty (e.g., which vegetable based meal is recommended to replace meat containing meals, and where should these vegetables be sourced from and what time of year) and furthermore makes no attempt to predict or understand the consequential changes in markets (consequential LCA) if dietary shift occurs.

Generally, we estimated larger proportional reductions in environmental impacts for meatless to meat-containing meals than previous comparisons of meatless and meat-containing diets [5,11,20,21,65–67]. For example, we found a greenhouse gas reduction of 58–77% for meals, where generally a 30–50% reduction of carbon footprint for entire vegetarian diets has been suggested [5,20]. We assume that we see a larger reduction for meals than that seen with diets because of two main reasons. First, non-vegetarian diets contain vegetarian meals such that the percent difference between the two diets would then be less than that per-meal. Second, impacts of meals do not contain out-of-mealtime snacks and beverages, which would increase the overall impact for both diets of similar magnitude, thus effectively lowering the percent difference between diets.

Several details were not included in this study that could influence both the proportional contribution of life cycle impacts of different stages and the total environmental impact estimations; for example, emissions from human waste were not included [68], and details regarding different food preparations were not considered [35]. Furthermore, any health impacts related to preparing and consuming foods were not included; for example, any increase in health risks due to particulate matter releases during cooking or for diets high in red or processed meat [69].

The meals compared were based on average consumptions and only scaled by mass to ensure no bias for meatless meals, which on average have less weight than meat-containing meals. A sensitivity study was performed to scale by calories and we found that because nutritional density was similar across meal types, scaling by calories offered the same scaling factor (different by 1%) as scaling by mass. We did not attempt to scale by protein, which we find more suitable for diet-level assessments instead of meal-level assessments. As a result of the study design, we made no attempt to ensure that meals were equivalent in nutritional value. Furthermore, we did not attempt to assess if the consumption of the average breakfast, lunch, and dinner provided a nutritionally complete meal. We therefore can make no recommendation for behavior change based on this study with respect to nutrition and health. From these results in this study, we can say that focusing on behavior change for shifting to meatless dinners can lead to the maximum benefits; about a two-fold improvement in absolute reductions in environmental impacts can be obtained from shifting to meatless dinners instead of breakfasts. Lunch meal occasions also offer an increased benefit to decrease environmental impacts when compared to breakfasts. Advantages found for dinner and thereafter lunch are due to the higher amount of meat consumed at lunch and dinner occasions in comparison to breakfasts.

The environmental impacts considered in this study were based on standardized life cycle impact assessment methodologies, and therefore did not consider emerging urgent issues, such as biodiversity loss due to land use change or effects on pollinators. In this way, this study's novelty is reflected mainly in the use of primary data for obtaining meal composition (instead of modelling or assuming meat replacements), and we make no attempt to improve or complement LCA methodology with respect to impact indicators.

## 5. Conclusions

This study used US nutritional dietary survey data to estimate the environmental impacts for the average meat-containing and meatless meals. These meals were not intended to be representative of what vegetarians and non-vegetarians consume, but of meals for the average American adult. This study can help inform industry where there are opportunities to lower environmental impacts of

meals (or products within meals). Interventions at the meal or product level tend to be more relevant for immediate commercial response than interventions targeted at the diet-level (e.g., increasing the number of vegetarians in a population), as prescribing a (new) dietary pattern is sensitive to social norms and context, and may not be relevant for the average consumer [70]. Meatless meals always had more than a 40% reduction in environmental impacts for any of the indicators (carbon footprint, water use, resource consumption, health impacts of pollution, and ecosystem quality); the biggest advantages were observed for dinner meal occasions due to the greater consumption of meat during dinner than other meals. Animal products in general (e.g., dairy) were major drivers of impacts across indicators, as well as for meatless meals, yet varied across meal occasions. Focusing on low-impact meat substitutes for dinners, and milk and yogurt substitutes for breakfasts, are examples of where commercial interventions (e.g., meat or dairy substitutes or alternative products) have the opportunity to lower impacts of meals in the US. Overall, the novelty of this study is that dietary survey data were used to obtain compositions of meatless and meat-containing meals.

**Supplementary Materials:** The following are available online at http://www.mdpi.com/2071-1050/11/22/6235/s1. Figure S1: Ranking of food items for the weight as consumed (averaged across meals) versus as produced in meat-containing meals, Figure S2: Contribution of food categories to the environmental impact of lunches, Figure S3: Contribution of food categories to the environmental impact of dinners, Table S1: Food categories and corresponding sub-categories including the proportional split for the different environmental inventory matches, Table S2: Percentage of food waste at retail and consumer levels for various food groups, adapted from Buzby et al., 2014, Table S3: Assumptions used to account for food preparation and clean-up. The same assumptions were used for all meals, scaled to the meal weight, Table S4: Average Consumption weight (grams) of meals and the production weight required for the environmental impact assessment, Table S5: Absolute values for environmental impact indicators of meal occasions for meatless and meat-containing meals, Table S6: Sensitivity analysis of the influence of various meat types.

**Author Contributions:** A.E. conceptualized and wrote this scientific article that was based on an ISO compliant study performed by Q.T., M.F., A.D.D., S.M. and J.D. The manuscript is based on an ISO compliant version available from morningstarfarms.com/about. An executive summary followed by a consumer-friendly tool based on the results of the study are also available at https://www.morningstarfarms.com/executive-summary.html.

**Funding:** This research was funded by the Kellogg Company.

**Conflicts of Interest:** The funders had no role in the design of the study; in the data collection, analyses, or interpretation of data; or in the writing of the manuscript. The funders approved the manuscript for publishing with minor comments that did not change the main findings or study design.

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
