# Peer review of "Comparing the Environmental Impacts of Meatless and Meat-Containing Meals in the United States"

_sustainability, doi:10.3390/su11226235_

Round 1

Reviewer 1 Report

The authors estimated the average environmental impacts of meatless versus meat containing meals in the US using data from NHANES. This objective is indeed novel, interesting, and could be an important contribution to the literature. I was excited to read this paper. At the same time, there are several methodological issues present that prevent me from recommending it for publication. I am concerned about the design of the study, and thus, the validity of the conclusions that can be drawn from it. Most concerning is the choice to use mass as a standardization basis for meals, followed closely by the lack of any assessment of uncertainty. The former is not defensible from a nutrition science perspective (see major comments), and the latter is not defensible from an LCA perspective. See my other major and minor comments below.

Major comments

·      Scaling/standardization method used for meals: I agree with your impulse to standardize across meals to make a fair comparison. Different age/gender/activity levels of respondents in the data mean that their caloric requirements vary substantially. The possibility for bias exists: middle aged women may be responsible for a greater proportion of the meatless meals, whereas young men may be responsible for a greater proportion of the meat-containing meals, for example. These groups have much different caloric requirements and likely caloric content of meals. That said, grams do not seem like a defensible standardization. In nutrition science, energy density is commonly used to standardize so that an isocaloric comparison can be made. While you rightly point out that using energy-based functional units can lead to misleading comparisons at the level of food items, they are appropriate if the unit of analysis is a meal or diet. See, for example, Rose et al. 2019, who compare the GHGs of individual diets from NHANES, per 1,000 kcal.   

·      Lack of management of discussion of uncertainty: uncertainty in any aspect of the data is not mentioned or addressed – this is a major oversight. Results are treated as point estimates. Sensitivity analyses are not performed. Thresholds for significant differences and quality of the underlying data are not discussed. We have no way of knowing how robust the results are without any discussion of uncertainty.

·      Handling of complex foods (mixtures): You make several assumptions regarding the composition of complex foods - of which there are likely hundreds in NHANES - such as frozen meals that contain meat. Datasets exist (FICD and FICRCD) to disaggregate NHANES (FNDDS) foods into their constituent commodities, but these were not used (see Heller et al. 2018, Blackstone et al. 2018 for examples of using these datasets). Given that the meal is your unit of analysis, that using individual consumption data is part of your pitch for the novelty of this approach, and that datasets exist to disaggregate complex NHANES foods, your approach seems like an oversimplification.

·      Inconsistent handling of beverages: It does not make sense to me to include some beverages (fruit juice and fluid milk) in meals, but exclude other beverages from meals. Beverages should be handled consistently.

·      Line 77: If your goal was to estimate the impact of a shift, an attributional LCA approach does not seem appropriate.  

·      Lines 242-245 (also in abstract): your study was not designed to assess a large scale shift in meals (see attributional comment above). I understand you are trying to give context here, but this is inappropriate.

Minor comments

·      Lines 24-27: it seems odd to lead with livestock production, when your paper is focused on meat consumption. I suggest leading with the environmental footprint of consumption, and then honing in on the contributions of animal source foods.

·      Line 33-34: add servings, and a citation, as a common comparison unit

·      Line 35: remove “per kilogram or kilocalorie (or any other unit-metric)”

·      Line 42-44: I disagree with the strength of this statement. Relative rankings, specifically within a food group or subgroup, are useful for consumer-facing guidance. If I’m in a grocery store, choosing what protein to serve for an entrée, then that relative ranking (per serving) matters. We just need to be very thoughtful about functional units when we compare individual food items in this way.  

·       Line 48: change “food” to “food consumption”

·      Lines 48-55: You should recognize recent scientific advances in this rea. Researchers have used datasets such as FoodAPS and NHANES to quantify the impacts of individual diets, average household food choices, and more. See: Boehm et al., 2018 and 2019, Heller et al. 2018, Rose et al., 2019.

·      Lines 45-63: The argument in this paragraph is difficult to follow. Please reorganize.

·      Lines 55-57: Subjectively choosing how substitutions are made, without including data on the likelihood of a substitution (e.g., through elasticities), is indeed a crude approach. At the same time, while your approach uses more data, I am not convinced it is a better approximation of a realistic substitution. First, no one actually chooses the “average meatless meal,” or the “average meat-based meal” – these are constructions. Second, the food preferences and socio-demographics of people choosing meatless versus meat-containing meals could be quite different.  

·      Line 70-73: This belongs in the methods section.

·      Line 98: Why were these food groups chosen, and where did the subcategories come from? Also “grains” as a group seems to be missing in the narrative. Additionally, your food groups in the narrative do not match what’s in the supplemental table. Where did the “grouped as” category come from in the supplemental table?

·      Line 102: More detail is needed on how you did this, either in the narrative, or in the SI.

·      Line 127-129: More detail is needed on how you did this, either in the narrative, or in the SI.

·      Line 130: Food waste is not assumed to be the same across meals, but the proportion of food wasted in each subgroup is. Please clarify.

·      Line 131-133: More detail is needed on how you did this, either in the narrative, or in the SI.

·      Line 133-134: What LCI data were used for waste management?

·      Lines 139-143: Be clearer on where the values you ended up using came from.

·      Lines 171-173: Please provide much more detail in the SI for the LCIs of your foreground processes. We should be able to see which unit processes were chosen for packaging, electricity, landfilling, etc. I suggest integrating some of this into the narrative (e.g., waste, mentioned above) as well.

·      Lines 171-173: what process/hierarchy was used to match LCI data on foods to the foods themselves? More detail is needed here. See also major comment on data quality and uncertainty above.

·      Line 174 – 176: There are seventeen midpoints and multiple endpoints – why do you only present a subset, and why this subset? You need to justify your impact scope.

·      Line 174-176: please also list the software and version that was used.

·      Results section: needs further quantification in the narrative in several places. Absolute results should be in the narrative. Logical flow could be improved: absolute results/meal comparison à contribution analyses.

·      Lines 201-206: these are methods not results.

·      Line 247: replace “reductions” with “differences”

·      Lines 274-276: It seems you are confusing diet-level and food-level analyses here. The impact/calorie issue related to the Tom study applied to their interpretations that compared food items per calories (ie. Lettuce and beef)

Reviewer 2 Report

This work provides a new approach to assessing trade-offs in environmental performance of meals with and without meat by using NHANES data to approximate actual consumption patterns. While there are some minor spelling or grammar issues (e.g. unnecessary comma on line 28, "compliant" spelled as "complient", the presentation is excellent. I have some suggestions for the research design, methodology, and results sections and areas where clarification is needed. 

In the introduction, line 27: Biomass conversion of cattle is probably the least efficient of all livestock. Are improvements not expected for more efficient animals such as chicken or pigs? This also does not address a common rebuttal that livestock can use otherwise marginal land not suitable for crops. 

In materials and methods, line 79-80: More detail is needed here regarding the functional unit - what was the exact definition as used in practice for conducting the assessment and was there any consideration for the other obligatory properties of a meal for its consumers (e.g. nutrient content, satiety, cultural concerns)? 

Line 96-97: It sounds like the meal is constructed based on national averages, but does that really avoid the problems outlined earlier regarding other LCA using constructed meals?

Line 110: Please clarify "quantity" as in mass unless it is something else.

Line 125: Needs clarification regarding other stages of the life cycle, and justification for not including them especially when plant-based meat analogues have the potential to have their environmental impacts significantly effected by processing stage impacts due to more extensive processing and low agricultural phase impacts.

Line 129-131: Food waste of meat is typically lower than that of produce, so a per group assessment based on national data should find a difference between meat containing and meatless meals for food waste.

Line 133-134: Did you consider an analysis including composting behavior? Does this assumption match actual disposal patterns?

Line 152: Actual packaging quantity may be lower than this value due to the assumption that there is a 50-50 split between snacks and beverages. Snacks and beverages will typically be in smaller packages than foods bought for meal preparation, and therefore may generate more packaging when considered on a weight basis.

Line 161: Double check your labeling for supplementary materials. This and other instances appear to be incorrectly labeled. S4 is regarding consumption weight calculations (it is also missing a "c" in "consumption") while S3 appears to have the expert assumptions. However, S3 also does not appear to include storage or water use.

Line 163: Please include a brief discussion of EIOLCA and how it differs from your description of LCA. 

Lines 186 and 194: Please double check again that the right SI table is listed, as these don't appear to match.

Line 194-195: My impression was that this article is intended to move away from relative comparisons of individual foods and diets or constructed meals, but it seems to do so anyway. Does the way the study was designed really avoid the problems described in the introduction? Provide some description or justification of how it does.

Line 210-211: Would you consider adding a "vegan" meal category as a comparison since even for meatless meals, animal-based products are still the major contributor to environmental impacts?

Line 242-244: This is helpful, but translating to another metric such as removing cars from the road etc. would help illustrate the point further for your audience.

Line 265-266: Can you provide an example of this? It is not clear that your results support such a statement. 

Line 268: Can you speculate as to why the difference is higher in this study?

Some other general concerns: This work appears to be mixing EIOLCA and process-based LCA, while not distinguishing between the two, unless I am mistaken and it only uses EIOLCA, in which case it is misrepresented by the description of LCA which is for more accurate for process-based LCA. 

There should also be a more clear delineation of system boundaries and description of what approach is used where. 

Table 1 appears to be the only part of the paper using kg of CO2 eq. to describe the difference in environmental impact from meals. It would be useful to also include the total kg CO2 eq. associated with each meal for meat containing and meatless in the main text. 

Reviewer 3 Report

I suggest to complete conclusions with a major discussion of results with international / european conclusions in similar case studies;

Reviewer 4 Report

The paper evaluates the environmental performance of meatless and meat containing meals in the USA across various metrics. The article is a first step in interpreting the environmental footprint of meals in the USA but requires large modifications throughout the manuscript, especially the environmental analysis, to be accepted to publishing. Please see my comments (both major and minor) below.

The introduction should emphasize that dietary shifts has emerged as an environmental mitigation strategy (with all relevant citations from the last few years), hence the motivation of this study. The authors mention the drawbacks of comparing individual food items, but it is less clear why meals is a better solution than full diets (e.g. more realistic?, easier to implement?, policy and interventions are easier to execute?). The introduction should be revised to give a broader perspective at the start and then elaborate more why meals is an important functional unit to explore compared to diets.

The real novelty is using realistic meals as detailed in NHANES. However the accuracy of the environmental analysis is unclear because LCA is used in a specific setting (farm practice) and climate and here it used as a national average. The authors mention they took ecoinvent values, but fail to mention what coefficients they used, did it include adjusting existing parameters to the USA, did they take many coefficients and then average them or any other meaningful information to help us understand how the environmental calculus was undertaken. The authors do not even mention this flaw in their discussion. Perhaps taking LCA values for food obtained from previous studies in the USA would be more accurate or taking average global values (e.g. Poore Nemeck Science 2018) and modeling it with the entailed global uncertainties. The values of the environmental impacts of the non-agricultural parts are also largely unclear (see my comments below).

Figures should be changed. Figure 1 is quite redundant, and a sentence detailing all lifecycle stages is sufficient. All figures should be modified to include colors – I found the black and white color codes difficult to follow. Figure 3 should be changed to “dinner” because these are the environmentally impactful meals, and thus a comparison can be more meaningful. Figure 4 should be scaled per impact (e.g. CO2 eq etc) not scaled to 100% of meat containing meal. This will enable a comparison of the true absolute values of impacts between the meals (breakfast vs dinner meat vs non meat etc) as well as with other studies and results. This will make table 1 mostly unnecessary.

It might have been interesting to model the environmental impacts of actual consumed meals (“as is” basis), meatless and meat containing meals without any scaling. This would have resulted in environmental differences stemming from differences in both quantities consumed and different per g impacts. Instead the authors normalize the meals on a per weight basis, which is odd. Comparing the meatless/meat meals should either be done on an “as is” basis or rather on an equal protein basis that can be meaningful from a “dietary shift” prospective.

The discussion lacks a comprehensive comparison to other LCA studies done in the USA. This would have helped to determine how close their “average” LCA values represent environmental impacts of actual USA consumed meals. A sensitivity analysis (say a Monte Carlo analysis based on different meal composition as extracted from NHANES and differing environmental parameters) would have been helpful to establish uncertainties. Surprisingly, the authors conclude that such sensitivities analysis (line 264-267) is outside the scope of this paper.  

In addition, because the analysis of the meals were done on a weight basis, the authors rightly conclude that they can’t infer on any recommendation apart from reducing meat (because of no nutritional consistency), leaving out any meaningful conclusions, policy or consumer recommendation as well as insights for future analysis.  

Minor comments:

The two first sentences in the abstract can be combined.

Line 16 unnecessary space

I think the example on lines 18-20 does not really give a feel for the magnitude. What is 0.2% GHG of total? I would use a more dramatic values, say 50% and see how that changes overall GHG emission. Or see my comment at the end.

Line 30-31 “environmental footprints of the food system” include many other aspects across the supply change including increased efficiency, reduction of waste etc. What is written is only from the demand side.

Please cite the Poore and Nemecek paper, Science 2018, line 34.

Line 43, I would also mention that the “ranking” is depended on specific agricultural practices and climates, and thus changes from one geographic setting to another might change its order.

Line 45 is a repeat but a better statement of line 43: “largely insufficient” should be softened, because relative ranking can indeed be informative, but sometimes can be misleading. But certainly not “largely insufficient”.

Line 105-107 is unclear. Why are there assumptions? I thought the consumption values are based on taking the averages of real meals?

Line 110 scaled by weight? If you want to compare differing meals I would use calories or protein not weight. This perhaps explains why the beverages take up the majority of weight.

Line 143 the environmental impacts of the non-agriculture stages (distribution/packaging) where than inferred from taking electricity grid impacts per energy unit? What about impacts from transport itself? Or pollution from manufacturing industries?

Line 195 – the sentence is unclear. Please rephrase.

Line 224 Unnecessary dot.

Line 270 Why are there such differences between this analysis and other studies. Did you compare the burdens per g you obtained to other studies? Maybe the non-meat diets have a different composition than other studies?

Line 310 “10% of footprint in Algeria”. Why Algeria? We have no idea what is the footprint there so “10%” doesn’t mean anything to most readers. Try to find meaningful values with which to compare your results. Use for example “Meatless Monday” estimates – giving up a meat containing meal per week across the entire USA. Use something that people can relate to, and compare for example to actual transportation costs in the USA or the like.

Round 2

Reviewer 4 Report

The revised paper is a great improvement to the original manuscript. I recommend accepting it. I have only small comments (see below)

minor comments:

line 157 - unnecessary "was used"

line 205 unnecessary space

The legend underneath the columns in figure 2 is very confusing, please separate the words. 

line 302 change to whereas

line 304 unnecessary not

line 364 if to of

line 380-384 break into sentences

line 386 erase the dot

Author Response

Dear reviewer, 

thank you for the positive feedback about the manuscript, and the close reading to spot the minor changes you recommend. Your review has helped improve the quality of the manuscript. We are grateful for having such an efficient, detailed, and complete peer-review process.

We have addressed all of the minor comments you have below in the manuscript and re-submitted for publication. 

Best, 

Alexi Ernstoff & other co-authors

Reviewer comments:

The revised paper is a great improvement to the original manuscript. I recommend accepting it. I have only small comments (see below)

minor comments:

line 157 - unnecessary "was used"

line 205 unnecessary space

The legend underneath the columns in figure 2 is very confusing, please separate the words. 

line 302 change to whereas

line 304 unnecessary not

line 364 if to of

line 380-384 break into sentences

line 386 erase the dot